# Mathematical modeling indicates that regulatory inhibition of CD8+ T cell cytotoxicity can limit efficacy of IL-15 immunotherapy in cases of high pre-treatment SIV viral load

Jonathan W. Cody[1], Amy L. Ellis-Connell[2], Shelby L. O'Connor[2], Elsje Pienaar[1,3]*

**1** Weldon School of Biomedical Engineering, Purdue University, West Lafayette, Indiana, United States of America, **2** Department of Pathology and Laboratory Medicine, University of Wisconsin-Madison, Madison, Wisconsin, United States of America, **3** Regenstrief Center for Healthcare Engineering, Purdue University, West Lafayette, Indiana, United States of America

* epienaar@purdue.edu

## Abstract

Immunotherapeutic cytokines can activate immune cells against cancers and chronic infections. N-803 is an IL-15 superagonist that expands CD8+ T cells and increases their cytotoxicity. N-803 also temporarily reduced viral load in a limited subset of non-human primates infected with simian immunodeficiency virus (SIV), a model of HIV. However, viral suppression has not been observed in all SIV cohorts and may depend on pre-treatment viral load and the corresponding effects on CD8+ T cells. Starting from an existing mechanistic mathematical model of N-803 immunotherapy of SIV, we develop a model that includes activation of SIV-specific and non-SIV-specific CD8+ T cells by antigen, inflammation, and N-803. Also included is a regulatory counter-response that inhibits CD8+ T cell proliferation and function, representing the effects of immune checkpoint molecules and immunosuppressive cells. We simultaneously calibrate the model to two separate SIV cohorts. The first cohort had low viral loads prior to treatment (≈3–4 log viral RNA copy equivalents (CEQ)/mL), and N-803 treatment transiently suppressed viral load. The second had higher pre-treatment viral loads (≈5–7 log CEQ/mL) and saw no consistent virus suppression with N-803. The mathematical model can replicate the viral and CD8+ T cell dynamics of both cohorts based on different pre-treatment viral loads and different levels of regulatory inhibition of CD8+ T cells due to those viral loads (i.e. initial conditions of model). Our predictions are validated by additional data from these and other SIV cohorts. While both cohorts had high numbers of activated SIV-specific CD8+ T cells in simulations, viral suppression was precluded in the high viral load cohort due to elevated inhibition of cytotoxicity. Thus, we mathematically demonstrate how the pre-treatment viral load can influence immunotherapeutic efficacy, highlighting the in vivo conditions and combination therapies that could maximize efficacy and improve treatment outcomes.

**Data Availability Statement:** All data and code required for replication, along with all data underlying figures, is available in a GitHub release

at <https://github.itap.purdue.edu/ElsjePienaarGroup/N803_SIV_Model/releases/tag/v2.0>.

**Funding:** This work was supported by seed funding from the Indiana Center for AIDS Research [https://indiana.pure.elsevier.com/en/projects/core-developmental] (to E.P.) and by National Institutes of Health [https://www.nih.gov/] award number R01AI108415 (to S.O.). The funders had no role in study design, data collection and analysis, decision to publish, or preparation of the manuscript.

**Competing interests:** The authors have declared that no competing interests exist.

## Author summary

Immunotherapy bolsters and redirects the immune system to fight chronic infections and cancers. However, the effectiveness of some immunotherapies may depend on the level of pre-treatment inflammation and the corresponding presence of regulatory cells and immune checkpoint molecules that normally function to prevent immune overreaction. Here, we consider two previously published cohorts of macaques who were given the immunotherapeutic N-803 to treat Simian Immunodeficiency Virus, an analog of Human Immunodeficiency Virus (HIV). One cohort had low viral loads before treatment, and N-803 temporarily suppressed viral loads. The second cohort had high viral loads that did not consistently decrease with N-803 treatment. In this work, we demonstrate with a mathematical model how these two distinct outcomes can arise due only to the different viral loads and the corresponding immune activation and regulatory response. In the model, we find that the key limiting factor is the direct inhibition of the cytotoxic action of immune cells by immune checkpoint molecules. This model indicates that simultaneous blockade of immune checkpoint molecules may be necessary for effective application of N-803 for the treatment of HIV. This and similar models can inform the design of such combination therapies for cancer and chronic infection.

## Introduction

Cytokines are the chemical messengers of the immune system, controlling cell division, apoptosis, differentiation, migration, and function [1,2]. Cytokines can be introduced therapeutically to activate the immune response against cancer [3] and chronic infections like human immunodeficiency virus (HIV) [4,5]. IL-15 is a cytokine that promotes proliferation and function of CD8[+] lymphocytes [6,7]. N-803 (ImmunityBio), formerly ALT-803, is an IL-15 superagonist that, when given to non-human primates (NHPs) infected with simian immunodeficiency virus (SIV), has expanded CD8[+] T cells [8–10], increased CD8[+] T cell cytotoxicity [9], reduced the number of SIV infected cells in the B-cell follicles [10], and transiently reduced plasma viral load [8]. SIV is a widely used animal model of HIV with similar, albeit accelerated, disease progression in rhesus macaques [11]. N-803 also increased CD8[+] T cell proliferation in humans participating in cancer trials [12,13] and HIV trials [14], and it increased cytotoxicity of human CD8[+] T cells in vitro [15]. While N-803 appears to be a promising treatment for HIV infection, the complex immunological responses to N-803 treatment remains unclear, and comparison across studies is complicated by variability in study designs. In this work, we aim to bridge experimental studies and characterize immune responses to N-803 treatment using a computational systems biology approach.

Mathematical models are a convenient complement to experimental studies and have been used to propose and evaluate hypotheses about HIV infection and the immune response for decades (reviewed in [16,17]). These models often take the form of ordinary differential equations built upon principles of reaction kinetics. Relevant applications of these models include investigating the role of regulatory T cells in HIV infection [18], predicting the benefit of anti-PD-L1 antibody therapy [19], and analyzing viral escape from the CD8[+] T cell response in a xenograft model of IL-15 therapy [20]. Our recent work [21] evaluated factors affecting the efficacy of N-803 in the previously mentioned cohort with transient viral suppression [8], finding that drug-induced immune inhibition, such as immune checkpoint molecule expression and regulatory T cells, could account for the observed loss of viral suppression with continued treatment.

Here we compared two cohorts of NHPs infected with SIV and treated with N-803 [8,9]. Cohort 1 [8] had a lower pre-treatment viral load ($\approx$3–4 CEQ/mL plasma) that was temporally suppressed with N-803 dosing, while Cohort 2 [9] had a higher pre-treatment viral load ($\approx$5–7 CEQ/mL plasma) that was not suppressed with N-803 dosing. The reason for the distinctly different outcomes between these two studies could be connected to the degree of pre-treatment viral control [9]. In chronic HIV, higher plasma viral load is associated with increased expression of immune checkpoint molecules on CD8$^+$ T cells [22,23] and higher regulatory T cell frequency [24,25]. Immune checkpoint molecules and regulatory T cells together form a counter-signal that limits a prolonged immune response by inhibiting CD8$^+$ T cell activation, proliferation, and cytotoxic function [26–28]. These regulatory mechanisms are an important limiting factor in cytokine monotherapy aiming to promote CD8$^+$ T cell control of cancer [3], including IL-15 therapy [29–31]. It is therefore possible that elevated regulatory factors in Cohort 2, induced by the higher viral load, precluded the viral suppression due to N-803 that was observed in Cohort 1. However, it is challenging to identify causal mechanisms from these experimental data alone.

In this study, we adapted and applied our computational model to simultaneously explain both the transient viral suppression in SIV Cohort 1 [8] and the lack of viral suppression in SIV Cohort 2 [9] after N-803 treatment. To this end, we fit one set of model constants to SIV viral load and CD8$^+$ T cell counts in both cohorts. By using the same model constants for both cohorts, we remove any inherent differences between them. We demonstrate how two very different viral responses to N-803 treatment can be observed within one system of equations if treatment is initiated from two different steady-states (viral loads, CD8$^+$ T cell counts, and corresponding regulatory inhibition of CD8$^+$ T cells). This work will contribute to our understanding of how the pre-treatment viral load can impact efficacy of immunotherapy.

## Methods

### Mathematical model

We followed the convention of using ordinary differential equation models of the dynamics of viral infection and immune response (reviewed in [16,17]). This is a lumped model that can be considered to convolve the dynamics in the blood and lymph. In this work, we updated our previous model of SIV infection and N-803 treatment [21] by separating the CD8$^+$ T cell pool according to activation and SIV-specificity. This change allows the level of CD8$^+$ T cell activation and regulatory inhibition to depend on both viral load and N-803, and it provides closer comparison to SIV-specific CD8$^+$ T cell cytotoxic marker expression [9]. Our differential equations (Eqs 1–11) track SIV-virions ($V$), resting ($S_0$) and active ($S_1$-$S_8$) SIV-specific CD8$^+$ T cells, resting ($N_0$) and active ($N_1$) non-SIV-specific CD8$^+$ T cells, injection site N-803 ($X$), bio-available N-803 ($C$), and a phenomenological representation of immune regulatory factors ($R_1$, $R_2$). Fig 1 schematically illustrates the model, all variables are defined in Table 1, and all constants are defined in Table 2. All data and code required to replicate this work (in MATLAB R2018b or later) can be download by following the link provided in S1 Text.

$$V' = qV - \left( \frac{g \sum_{i=1}^{8} S_i}{1 + \lambda R_2} \right) V \qquad (1)$$

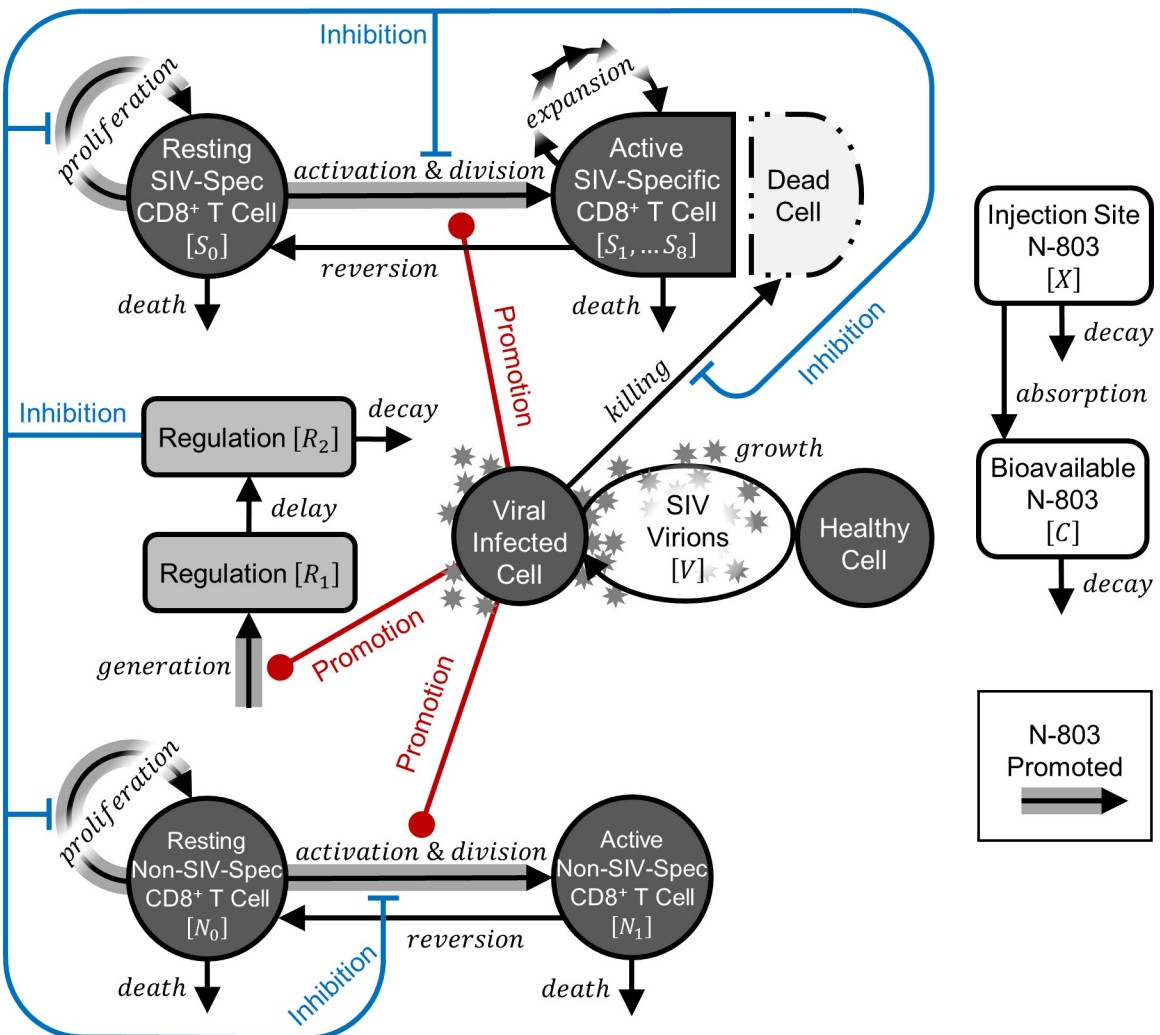

**Fig 1. Mathematical model of SIV infection and N-803 treatment.** The ordinary differential equation model was adapted from previous work [21]. SIV-virions ($V$, Eq 1) grow exponentially and activate resting SIV-specific ($S_0$, Eq 2) and non-SIV-specific ($N_0$, Eq 6) CD8+ T cells. Activated SIV-specific ($S_1$-$S_8$, Eqs 3–5) CD8+ T cells undergo programmed expansion (indicated by the chain of arrows), suppress the infection, and revert to a resting state. Activated non-SIV-specific ($N_1$, Eq 7) CD8+ T cells divide only once before reversion. Activation by virus also promotes a phenomenological regulatory response ($R_{1,2}$, Eqs 10 and 11) that represents inhibition of CD8+ T cell proliferation, activation, and function by immunosuppressive cells and immune checkpoint molecules. Injection site N-803 ($X$, Eq 8) is absorbed and becomes bioavailable N-803 ($C$, Eq 9), where it promotes CD8+ T cell proliferation and activation, as well as regulation.

$$S_0' = \frac{pS_0}{1+\varphi R_2}\left(\frac{S_{50}}{S_{50}+\sum_{i=0}^{8}S_i}\right)\left(1+\frac{\rho C}{C_{50}+C}\right) - dS_0 - \frac{a_S S_0}{1+\zeta_S R_2}\left(\frac{V}{V+V_{50,S}}\right)$$
$$\times \left(1+\frac{\alpha_S C}{C_{50}+C}\right) + m_S S_8 \tag{2}$$

$$S_1' = 2\frac{a_S S_0}{1+\zeta_S R_2}\left(\frac{V}{V+V_{50,S}}\right)\left(1+\frac{\alpha_S C}{C_{50}+C}\right) - d_A S_1 - p_A S_1 \tag{3}$$

**Table 1. Model variables.**

| Variable | Symbol | Initial Value [a] | Units | Ref. [c] |
|---|---|---|---|---|
| SIV virions (Cohort #1) | $V$ | (1, 15) | CEQ/μL | [8] |
| SIV virions (Cohort #2) | $V$ | (100, 4000) | CEQ/μL | [9] |
| Resting SIV-specific CD8$^+$ T cells | $S_0$ | Calculated [b] | #/μL | |
| Active SIV-specific CD8$^+$ T cells | $S_1$-$S_8$ | Calculated [b] | #/μL | |
| Resting non-SIV-specific CD8$^+$ T cells | $N_0$ | Calculated [b] | #/μL | |
| Active non-SIV-specific CD8$^+$ T cells | $N_1$ | Calculated [b] | #/μL | |
| Absorption site N-803 | $X$ | 880 | pmol/kg | [32] |
| Bioavailable N-803 | $C$ | 0 | pM | |
| Regulation (Cohort #1) | $R_1, R_2$ | 1 | (normalized) | |
| Regulation (Cohort #2) | $R_1, R_2$ | Calculated [b] | (normalized) | |
| Total CD8$^+$ T cells [b] | | (200, 1000) | #/μL | [8,9] |
| SIV-specific frequency in CD8$^+$ T cells [b] | | (0.03, 0.3) | | [33–35] |

[a] During the parameter fitting process, initial conditions were allowed to be different for each cohort, with the exception of N-803 initial conditions ($X$, $C$). Values shown in pairs indicate allowed ranges for the parameter fit, while single values were held fixed and not fitted. Allowed ranges were the same for both cohorts, unless listed separately.

[b] Some initial conditions were calculated by assuming a steady-state for both cohorts prior to treatment. The initial total CD8$^+$ T cells and the initial frequency of SIV-specific cells within total CD8$^+$ T cells were fitted for each cohort within ranges shown and used in calculation of initial conditions for CD8$^+$ T cell subgroups ($S_1$-$S_8$, $N_0$, $N_1$). See S1 Text for details on this calculation.

[c] See S1 Text for a discussion of how initial conditions were obtained from given references.

$$S'_i = 2p_A S_{i-1} - d_A S_i - p_A S_i \quad \text{where} \quad i \in 2, 3, \ldots 7 \tag{4}$$

$$S'_8 = 2p_A S_7 - d_A S_8 - m_S S_8 \tag{5}$$

$$N'_0 = \frac{pN_0}{1 + \varphi R_2}\left(\frac{N_{50}}{N_{50} + N_0 + N_1}\right)\left(1 + \frac{\rho C}{C_{50} + C}\right) - dN_0 - \frac{a_N N_0}{1 + \zeta_N R_2}\left(\frac{V}{V + V_{50,N}} + \frac{\alpha_N C}{C_{50} + C}\right) + m_N N_1 \tag{6}$$

$$N'_1 = 2\frac{a_N N_0}{1 + \zeta_N R_2}\left(\frac{V}{V + V_{50,N}} + \frac{\alpha_N C}{C_{50} + C}\right) - d_A N_1 - m_N N_1 \tag{7}$$

$$X' = -k_a X \tag{8}$$

$$C' = k_a\left(\frac{F}{v_d}\right)X - k_e C \tag{9}$$

$$R'_1 = s_S\left(\frac{V}{V + V_{50,S}}\right)\left(1 + \frac{\alpha_S C}{C_{50} + C}\right) + s_N\left(\frac{V}{V + V_{50,N}} + \frac{\alpha_N C}{C_{50} + C}\right) - d_R R_1 \tag{10}$$

$$R'_2 = d_R R_1 - d_R R_2 \tag{11}$$

**Table 2. Model constants.**

| Constant [a] | Symbol | Value [b] | Units | Ref. [d] |
|---|---|---|---|---|
| Viral growth rate constant | $q$ | (0.01, 1) | /day | [36] |
| Killing rate constant | $g$ | Calculated [c] | μL/#/day | |
| Activation rate constants (ST8,NT8) | $a_S, a_N$ | Calculated [c] | /day | |
| SIV concentration for 50% activation (ST8, NT8) | $V_{50,S}, V_{50,N}$ | (10, 1000) | CEQ/μL | |
| Reversion rate constant (ST8) | $m_S$ | (0.004, 0.04) | /day | [37] |
| Reversion rate constant (NT8) | $m_N$ | (0.04, 0.4) | /day | |
| Concentration for 50% proliferation (ST8, NT8) | $S_{50}, N_{50}$ | Calculated [c] | #/μL | |
| Sum of 50% proliferation concentrations [c] | $S_{50}+N_{50}$ | (300, 3000) | #/μL | |
| Proliferation rate constant (resting T8) | $p$ | (0.01, 0.1) | /day | |
| Proliferation rate constant (active ST8) | $p_A$ | (1, 4) | /day | [37,38] |
| Death rate constant (resting T8) | $d$ | (0.01, 0.05) | /day | [39] |
| Death rate constant (active T8) | $d_A$ | (0.1, 0.5) | /day | [37,38] |
| N-803 absorption rate constant | $k_a$ | 0.80 | /day | [40] |
| N-803 clearance rate constant | $k_e$ | 2.1 | /day | [15] |
| N-803 volume of distribution / bioavailability [c] | $d_d/F$ | 1.3 | L/kg | [15,40] |
| N-803 concentration for 50% effect | $C_{50}$ | (1, 1000) | pM | [10,15] |
| N-803 induced expansion rate (resting T8) [c] | $\rho \cdot p$ | (0.1, 2) | /day | [41] |
| N-803 stimulation factor for activation (ST8) | $\alpha_S$ | (0.1, 1000) | | |
| N-803 stimulation factor for activation (NT8) | $\alpha_N$ | (0.01, 100) | | |
| Regulation generation rate (ST8) | $s_S$ | Calculated [c] | /day | |
| Regulation generation rate ratio (NT8/ST8) [c] | $s_N/s_S$ | (0.1, 100) | | |
| Regulation delay rate constant | $d_R$ | (0.05, 1) | /day | |
| Regulation factor for killing | $\lambda$ | Calculated [c] | | |
| Regulation factor for activation (ST8,NT8) | $\zeta_S, \zeta_N$ | Calculated [c] | | |
| Regulation factor for proliferation | $\varphi$ | (0.01, 100) | | |

[a] T8 refers to CD8+ T cells, ST8 refers to SIV-Specific CD8+ T cells, and NT8 refers to non-SIV-specific CD8+ T cells.

[b] During the parameter fitting process, the same values for constants were used when simulating both cohorts. Values shown in pairs indicate allowed ranges for the parameter fit, while single values were held fixed and not fitted.

[c] Some constants are calculated by assuming a steady-state for both cohorts prior to treatment, and some constants were calculated from fitted combinations. See S1 Text for details on these calculations.

[d] See S1 Text for a discussion of how constants were obtained from given references. Ranges without reference typically have no experimental analogue and are sampled within a mathematically relevant range informed by model development.

## Virus

SIV virions ($V$, Eq 1) grow exponentially (rate constant $q$) in the absence of CD8+ T cells. Viral growth is controlled by targeting of infected cells by active SIV-specific CD8+ T cells (2nd order rate constant $g$). Note that this simplified infection model can be obtained from a model with healthy cells and infected cells by assuming constant healthy cells and a quasi-steady-state for virions [21,42,43]. The former assumption follows from CD4+ T cell data collected with our training data that showed very little change in CD4+ T cell numbers throughout treatment [8] and from the low frequency of infected cells in chronic HIV [44]. The latter assumption is justified by considering how quickly SIV virions are cleared from the blood [45]. We also did not explicitly consider latent infection. The frequency of latently infected cells in chronic HIV infection is small [46], and our Cohort 1 saw only brief periods of viral control [8].

## CD8$^+$ T cells

Resting CD8$^+$ T cells ($S_0$,$N_0$, Eqs 2 and 6) proliferate and die with shared rate constants ($p$,$d$ respectively), reflecting maintenance of this population by self-renewal [47,48]. Density-dependence ($S_{50,50}$ terms) is included for stability and reflects competition over space and cytokines [49]. CD8$^+$ T cells are activated and divide (rate constants $a_{S,N}$ Eqs 2,3,6 and 7) based on saturating functions of viral load ($V_{50,S}$,$V_{50,N}$ terms). Active SIV-specific CD8$^+$ T cells ($S_1$-$S_8$, Eqs 3–5, representing 8 generations of cells) undergo expansion, which is modeled as a chain of 7 additional divisions (rate constant, $p_A$), followed by reversion to a resting state ($m_S$). Active cells also die faster than resting cells ($d_A$). This framework is adapted from models of CD8$^+$ T cell clonal expansion [37,38]. There is also substantial non-specific activation of CD8$^+$ T cells in HIV infection [50–52]. Here, active non-SIV-specific CD8$^+$ T cells ($N_1$, Eq 7) are modeled similarly to SIV-specific cells, but do not undergo large, programmed expansion. While CD8$^+$ T cells are capable of non-specific cytotoxicity [53], including a killing for non-SIV-specific cells did not improve the fit to the non-human primate data. This and other variations of the formulation described above are compared using Akaike Information Criteria (Figs A-E in S1 Text). While simpler formulations can yield similar qualitative model behavior, the chosen model provided the best balance between agreement with key features of the calibration data and model complexity.

## Pharmacokinetics and pharmacodynamics

Following a standard pharmacokinetic model for subcutaneous dosing [54], N-803 ($X$, Eq 8) is absorbed from the site of injection (rate constant $k_a$). A fraction ($F$) that is absorbed becomes a bioavailable concentration ($C$, Eq 9) over a volume of distribution ($d_d$), before being eliminated (rate constant $k_e$). In our pharmacodynamic model, N-803 increases CD8$^+$ T cell proliferation [8,10] and cytotoxic function [9,15] based on multiplicative functions (Eqs 2,3,6 and 7) that reach half-saturation at a single concentration ($C_{50}$). Note that drug and viral promotion of non-specific activation are additive with each other (Eqs 6 and 7), reflecting how IL-15 promotes non-specific activation [51]. In contrast, programmed proliferation of SIV-specific CD8$^+$ T cells requires the virus (Eqs 2 and 3) but is accelerated by IL-15 [55].

## Immune regulation

Regulatory T cells and inhibitory molecules function together to prevent overreaction of the cytotoxic response (reviewed in [26–28]). We model the inhibitory effect phenomenologically with dimensionless 'regulation' variables ($R_{1,2}$, Eqs 10 and 11). Regulation is generated based on the SIV-specific and non-specific CD8$^+$ T cell activation signals from the virus and N-803 (scaled by $s_S$,$s_N$) after a delay (rate constant $d_R$), thus accounting for the effect of IL-15 on these regulatory pathways [8,56–59]. Immune regulation acts by inhibiting CD8$^+$ T cell proliferation (strength governed by $\varphi$, Eqs 2 and 6), activation ($\zeta_S$,$\zeta_N$, Eqs 2,3,6 and 7), and killing of infected cells ($\lambda$, Eq 1).

## Data summary

Our mathematical model was simultaneously calibrated to two non-human primate (NHP) cohorts [8,9]. These cohorts were each given N-803 during the chronic stage of a simian immunodeficiency virus (SIV) infection (Table 3). Plasma viral load, measured as viral RNA copy equivalents (CEQ), and peripheral blood CD8$^+$ T cells from both cohorts were used to calibrate the model. We distinguish the cohorts by their viral load at the time of treatment, though there are other noteworthy differences. The first cohort [8], consisted of four rhesus

**Table 3. Summary of SIV Cohorts used for calibration.**

| Cohort | #1 (Low Viral Load) [8] | #2 (High Viral Load) [9] |
|---|---|---|
| NHP species | Indian rhesus macaques (*Macaca mulatta*) | Indian rhesus macaques (*Macaca mulatta*) |
| Size | 4 | 12* |
| Number of data points | For SIV CEQ in plasma: n = 135<br>For CD8+ T cells in blood: n = 93 | For SIV CEQ in plasma:<br>n = 217<br>For CD8+ T cells in blood:<br>n = 108 |
| MHC Expression | *Mamu-B*08+* (3)<br>*Mamu-A*01+* (1) | *Mamu-A*01+* |
| Vaccination | All vaccinated pre-infection [60] | 7 of 12 vaccinated pre-infection |
| Infecting virus | *SIVmac239* | *SIVmac239M* [62] |
| Control history | Temporary viral control<br>(<50 CEQ/mL) [60] | No history of viral control |
| Viral load at start of N-803 treatment | ≈3–4 log CEQ/mL | ≈5–7 log CEQ/mL |
| Time post-infection of N-803 treatment | 1.5–3 years | 6 months |
| N-803 delivery | 0.1 mg/kg subcutaneous dose<br>Once per week for four weeks, one week pause, once per week for four weeks, 29 week pause, once per week for four weeks | 0.1 mg/kg subcutaneous dose<br>Once every two weeks for 3 doses total |

* There were 15 animals in this study [9], but only 12 are used as model calibration data. Three are excluded from calibration due to their inconclusive viral response resulting from initial viral loads near or below the detection limit.

macaques, infected with *SIVmac239* and having low but detectable viral loads (≈3–4 log CEQ/mL at the time of treatment). This cohort had received vaccination against SIV prior to infection and had temporarily maintained undetectable SIV (<50 CEQ/mL) [60], and three of these macaques expressed an MHC allele associated with SIV control [61]. In the second cohort [9], fifteen rhesus macaques were infected with *SIVmac239M*, which is a barcoded *SIVmac239* [62], and ten of these also received some type of prior vaccination. Of the fifteen NHPs, twelve had high viral loads (≈5–7 log CEQ/mL at the time of treatment). Three had viral loads near or below the detection limit, so as to make the viral load response to N-803 inconclusive for these subjects. Thus, only the twelve high viral load subjects were included in model calibration. We also compared our model predictions to measurements of Granzyme-B, a marker of CD8+ T cell cytotoxic function, in our calibration cohorts [9] and to data from additional healthy and SIV-infected NHP cohorts given intravenously administered N-803 [10].

## Parameter estimation

The goal is to use our mathematical model (Eqs 1–11) to replicate changes in viral load and CD8+ T cell counts in two NHP cohorts (Table 3) [8,9] using different initial conditions for each cohort (Table 1), yet the same set of model constants (Table 2). As such, the fitting process can be split into two levels. The top level (Fig 2, left side) is a series of algorithms that generate parameter sets that best reproduce the data from both cohorts for the given model structure. In this context, a parameter set is a set of numerical values, representing initials, constants, or combinations thereof, given in Tables 1 and 2. The bottom level (Fig 2, right side) is the function that takes a parameter set and returns a model cost, which is a measure of error with respect to data (Eq 12). The parameter set is first used to calculate two sets of initial

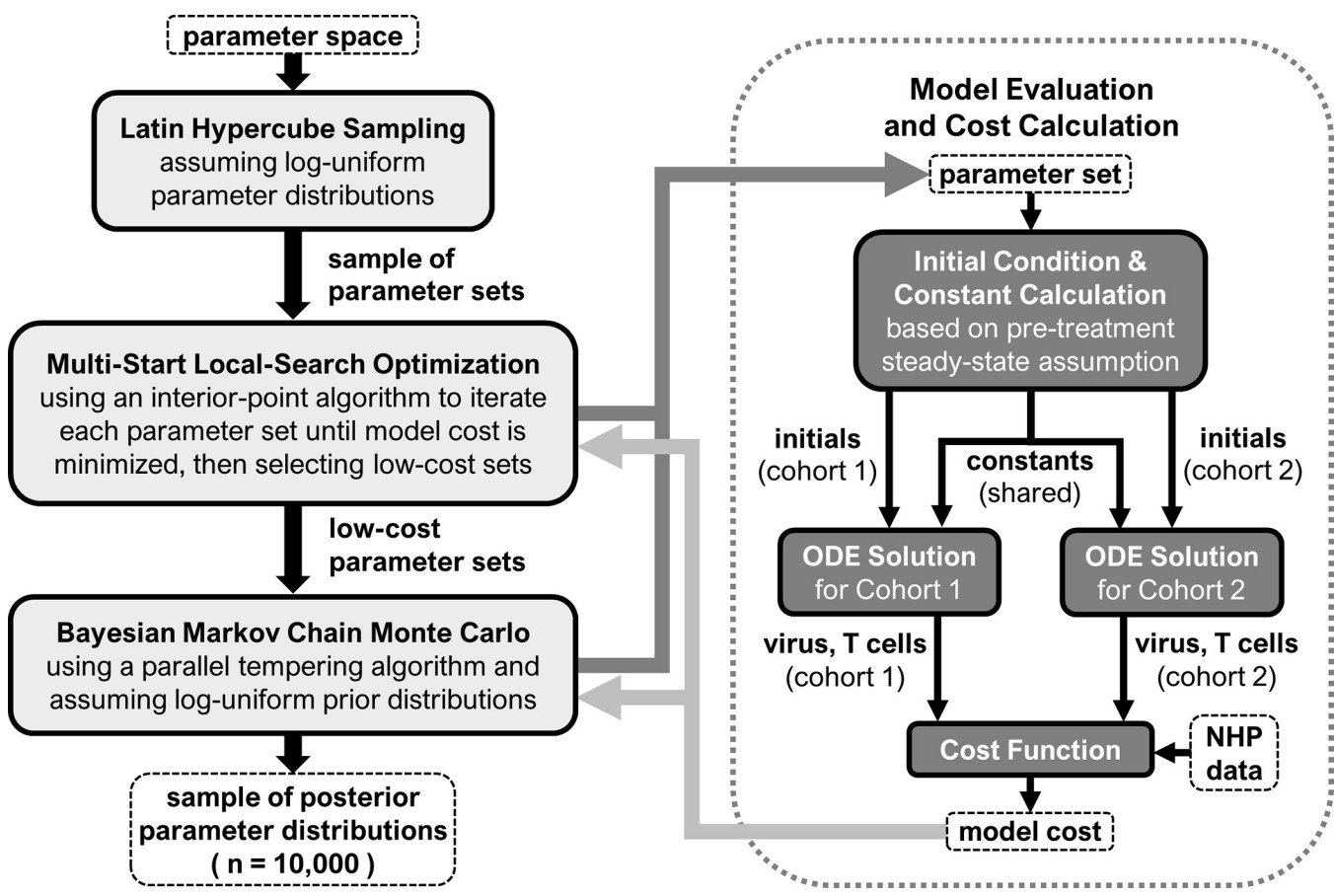

**Fig 2. Parameter fitting process.** Described is the process for obtaining 10,000 parameter sets for the mathematical model of N-803 treatment of SIV (Eqs 1–11, Fig 1) that replicate SIV viral load and CD8⁺ T cell counts from two non-human primate cohorts (Table 3). The left side describes the top-level parameter fitting algorithms. The right side describes the bottom-level calculation of model costs by solving the model from different initial conditions corresponding to each cohort and a shared set of model constants.

conditions, distinct for each cohort, and one set of constants, shared by both cohorts, based on the assumption that both cohorts are at a steady-state prior to treatment. Details on this calculation are provided in S1 Text. Details on the various steps in Fig 2 are described below.

### Parameter fitting algorithms

The top-level parameter fitting process, implemented in MATLAB version R2018b (Mathworks), is summarized in Fig 2 (left side). First, a Latin hypercube sample [63] of initial conditions and model constants was generated by assuming log-uniform distributions bounded by the ranges in Tables 1 and 2. An interior-point algorithm [64] was used to iterate each parameter set and minimize a model cost function (Eq 12). The five parameter sets with the lowest model cost were then used to initialize a parallel tempering Markov chain Monte Carlo algorithm [65] (*PESTO* toolbox [66]). This algorithm generated a sample (10,000 parameter sets) from the Bayesian posterior distribution of parameter values, again assuming log-uniform prior distributions for parameters. Model output figures were created by running the model for all 10,000 sets and then taking the inner 95% of the model predictions.

### Equation solving algorithms

Ordinary differential equations (Eqs 1–11) were solved using MATLAB's stiff equation solvers, with settings dependent on the task. For parameter estimation and uncertainty quantification, we used *ode23s* with 1% relative tolerance and 0.001 absolute tolerance. Tolerances were chosen by comparing model outputs over the time interval of the NHP data, with 1% relative tolerance yielding no appreciable difference from tighter tolerances. In *ode23s*, absolute tolerance is the lowest model output value for which error is considered, which we set based on a trial calibration of the model. For plotting, we used *ode15s* with default error tolerances (0.1% relative tolerance, $10^{-6}$ absolute tolerance). In both cases, an analytical Jacobian function was provided to accelerate solutions and improve calibration outcomes.

### Cost function

The goal of this work is to demonstrate how distinct responses to immune therapy can arise when therapy is given at different initial states of the immune system (different levels of virus, inflammation, regulation, etc.). There are four response variables that we compare to experimental data, SIV virus and $CD8^+$ T cells from each of the two cohorts. We assume independent, identical, and normally distributed error for each response variable, and we neglect any error covariance between virus and $CD8^+$ T cells. This allows the collective negative log likelihood (NLL) to be defined as the sum of the NLL of each of the four response variables (Eq 12).

$$NLL = \sum_{i=1}^{4} \frac{n_i}{2}[1 + \ln(MSE_i)] \qquad (12)$$

Eq 12 is formulated by the concentrated likelihood method [67] and is a function of mean squared error (MSE) and of the number of data points (Table 3) for the response variable. In order to balance the cost between response variables, we log transform viral load (in CEQ/mL) and we convert $CD8^+$ T cell concentrations (to cells/nL) prior to evaluating Eq 12.

## Results

### Pre-treatment state of immune system can determine outcome of immune therapy

We simultaneously fitted our model to two SIV cohorts given N-803 (Fig 3), with Cohort 1 having a low viral load at the start of treatment ($\approx$3–4 log CEQ/mL) and Cohort 2 having a high viral load ($\approx$5–7 log CEQ/mL). Identical model constants were used for both cohorts, requiring that their distinct initial conditions lead to distinct responses to N-803 in the mathematical model. Both of these cohorts are from previous studies and are summarized in Table 3. Fig 3A compares the simulated plasma viremia (Bayesian 95% credible interval) to the NHP data for Cohort 1 [8]. Following treatment initiation, the simulated viremia drops from an initial value of 3.5–3.8 log CEQ/mL down to 2.0–2.5 log CEQ/mL within the first two weeks, compared to a drop from ~3–4 log CEQ/mL to <2 log CEQ/mL in NHP Cohort 1 (each individual dropping below the detection limit of the assay). The simulated viral load then partially rebounded to 2.9–3.3 log CEQ/mL, compared to 2.0–3.5 log CEQ/mL in the experimental data, by the start of the second cycle (week 5). There was minimal response to the second cycle of doses in both the simulations and experimental data. In response to the third treatment cycle (week 37), the simulated viremia dropped to 2.2–2.7 log CEQ/mL, with the data dropping to <2–2.8 log CEQ/mL, depending on the individual. Thus, there was a partial recovery in the efficacy of viral suppression after the long treatment break.

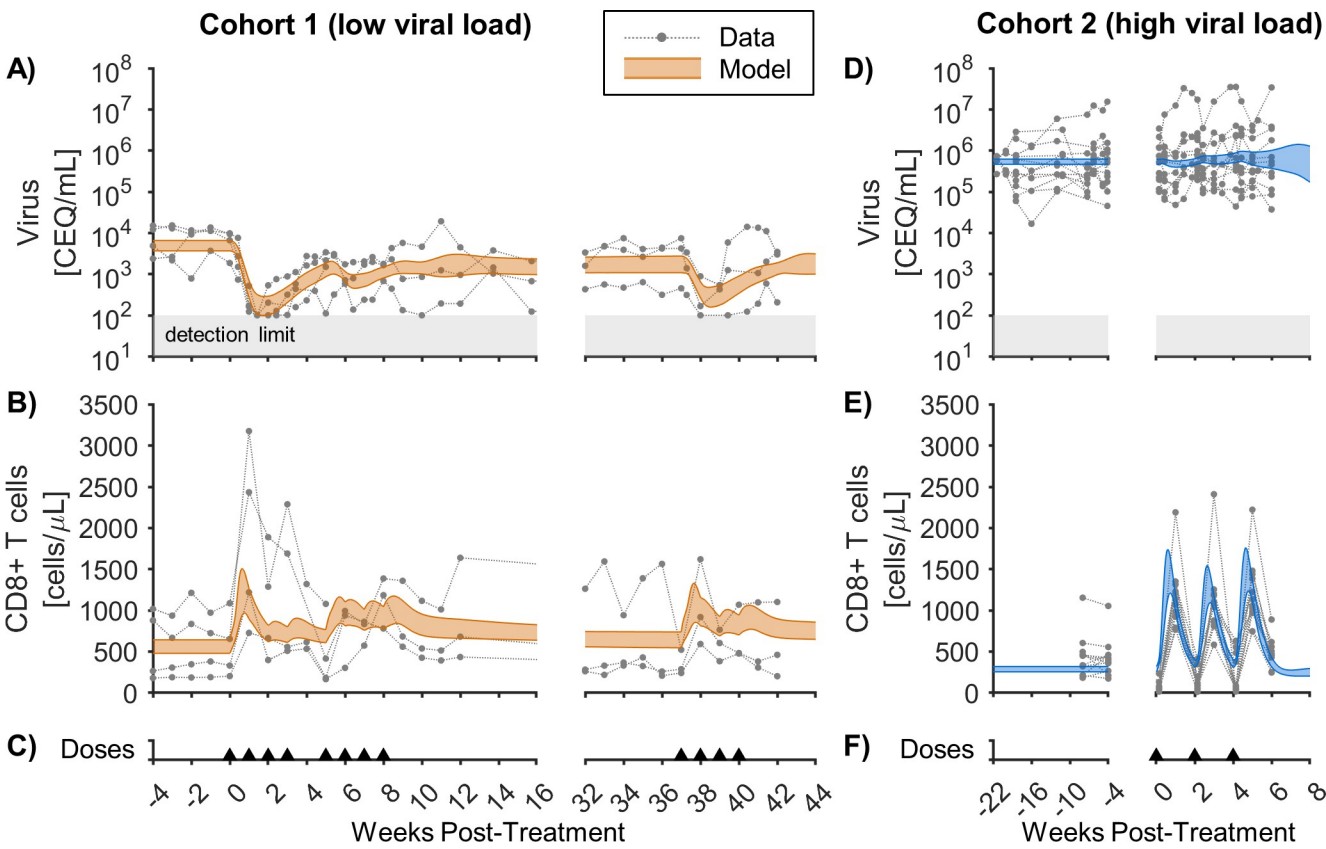

**Fig 3. Model calibration to N-803-treated SIV cohorts with low and high viral loads.** The model was calibrated to (A,D) virus in the plasma and (B,E) CD8[+] T cells in the peripheral blood from two different Simian Immunodeficiency Virus (SIV) cohorts. Cohort 1 (A,B) had a low viral load (≈3–4 log CEQ/mL) at the start of treatment [8], while Cohort 2 (D,E) had a high viral load (≈5–7 log CEQ/mL) [9]. The orange and blue shaded regions correspond to the Bayesian 95% credible interval of the mathematical model. The gray shaded region indicates the lower limit of detection for the viral assay (100 CEQ/mL). Panels (C,F) show timing of 0.1 mg/kg subcutaneous doses of N-803. Corresponding parameter distributions are shown in S1 Fig, and comparison of the model to the mean data of each cohort is shown in S2 Fig.

Fig 3D similarly compares the simulated plasma viremia (initially 5.6–5.8 log CEQ/mL) to the measured viremia for Cohort 2 (≈5–7 log CEQ/mL) [9]. In contrast to Cohort 1, viremia did not consistently decrease in response to treatment in Cohort 2. Note that, in Cohort 1, viral load dropped precipitously (in both simulation and data) before the second dose is given, so the dosing regimens alone do not account for the differences in viral dynamics.

In contrast to viral load, CD8[+] T cells increased in both cohorts in response to the first dose (Fig 3B and 3E). CD8[+] T cells expanded from 470–640 cell/μL to 960–1500 cell/μL in the simulations for Cohort 1 and from 250–320 cell/μL to 1200–1700 cell/μL in the simulations for Cohort 2. In Cohort 1 (Fig 3B), the simulated CD8[+] T cells contracted after the initial expansion, increasing again after the 1 week break in doses. This is qualitatively consistent with the data, though there was some variability in the data between individuals. In contrast, Cohort 2 (Fig 3E), with doses at 2-week intervals, showed a more consistent expansion and contraction of cells with each dose. It is worth noting that the sampling timeline is different for the two groups with Cohort 2 having data collected 1 day after each dose, and Cohort 1 having data collected immediately before each dose. A drop in peripheral blood CD8[+] T cells in Cohort 2 was observed one day after each dose. This could be due to migration of CD8[+] T cells from the blood to the lymph tissue [68] or mucosal tissue [69], which our model does not explicitly consider.

Taken together, our simultaneous calibration to two NHP cohorts, with shared model constants, supports the theory that pre-treatment viral load can account for starkly different viremia dynamics in response to immune therapy, despite similar CD8$^+$ T cell expansion. To further support this finding, we calibrated our model to each cohort separately, to evaluate if allowing constants to be different between the two cohorts we can achieve better model fits. Indeed, the quality of model fit did not significantly improve when fitting constants to each cohort separately (S3 Fig).

### The model replicated concurrent CD8$^+$ T cell cytotoxicity data

To test our model, we compared our simulated frequency of cytotoxic cells among SIV-specific CD8$^+$ T cells ($\Sigma S_{1...8}/\Sigma S_{0...8}$) to experimental data that were not used in calibration (Fig 4). In the experimental data, expression of Granzyme-B among SIV-specific effector memory (EM) CD8$^+$ T cells is shown [9], where Granzyme-B is a marker of cytotoxic function [70]. In our simulations, Cohort 1 (Fig 4A) started with a lower cytotoxic cell frequency, 7–21%, compared to 47–70% in Cohort 2 (Fig 4C). Pre-treatment cytotoxic frequency was similarly distinguished between the experimental cohorts, which show a frequency of 16–23% in Cohort 1 compared to 22–75% in Cohort 2. By the third day after N-803 administration, the simulated cytotoxic frequency was high for both cohorts: 72–90% in Cohort 1 and 97–99% in Cohort 2. This is compared to the experimental data showing 46–78% for Cohort 1 and 55–94% for Cohort 2. At day 7, model predicted cytotoxic frequency was similar to day 3 for both cohorts, while measured Granzyme-B frequency lowered to 45–48% in Cohort 1 and 22–79% for Cohort 2. Simulation and data were still in qualitative agreement in that SIV-specific cytotoxicity started lower in Cohort 1 than in Cohort 2 but N-803 moved the cohorts closer together. Our model also corresponded qualitatively with an independent dataset measuring viral load and CD8$^+$ T cells after intravenous dosing in SIV-infected and SIV-naïve NHPs [10] (S4 Fig). Thus, the

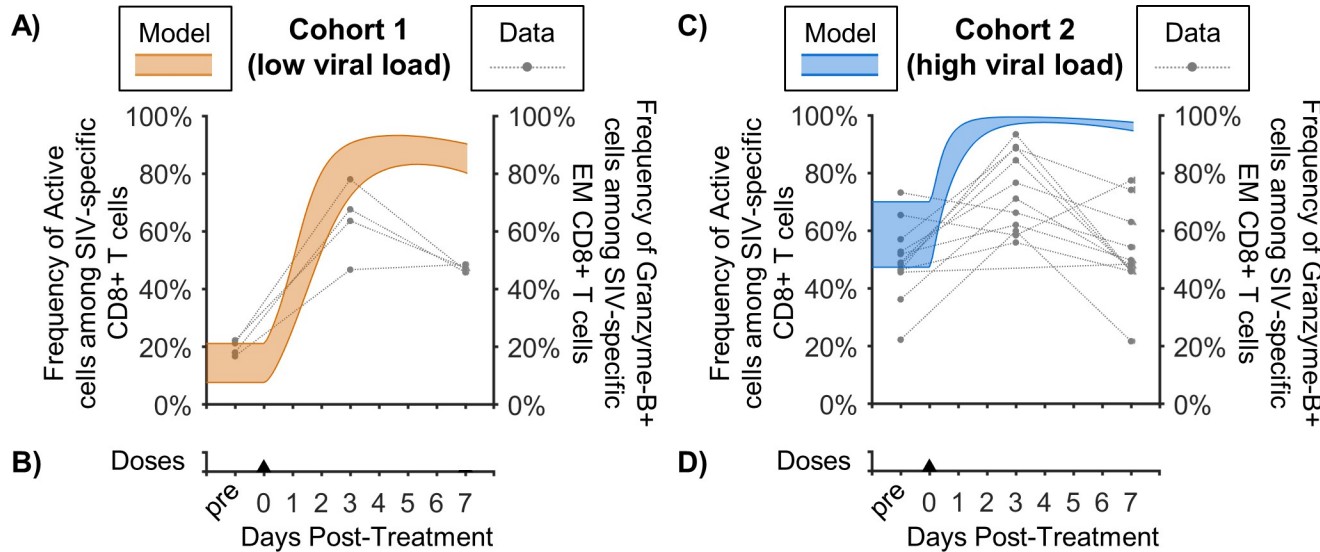

**Fig 4. Comparison of model predicted cytotoxic frequency among SIV-specific CD8$^+$ cells.** Shown are model predictions for the frequency of cytotoxic cells among SIV-specific CD8$^+$ T cells (Bayesian 95% credible interval) for Cohort 1 (A) and Cohort 2 (C). In the model, this is the ratio of active SIV-specific CD8+ T cells to total SIV-specific CD8+ T cells ($\Sigma S_{1...8}/\Sigma S_{0...8}$). The model is compared to the frequency of Granzyme-B$^+$ cells among SIV-specific effector memory (EM) CD8$^+$ T cells [9] taken before treatment and at days 3 and 7 after the first dose of N-803. These cells were specific to either Gag$_{181-189}$CM9 or Nef$_{137-146}$RL10 epitopes, depending on primate MHC expression. Panels (B,D) show timing of 0.1 mg/kg subcutaneous doses of N-803.

model was supported with both supplementary data from the cohorts of interest and from other NHP cohorts.

## Regulatory inhibition of CD8$^+$ T cell cytotoxicity precludes suppression in the high viral load cohort

Since our calibrated simulations indicated that pre-treatment plasma viral load can affect treatment response, we next took advantage of our computational approach and quantitatively uncoupled the contribution of individual mechanisms to the simulated viral and immune dynamics. Fig 5 shows a selection of terms from Eqs 1–11 that are most relevant to explaining the different dynamics observed in the two cohorts. Since we were most concerned with comparing the cohorts, all values in Fig 5 are normalized to the corresponding Cohort 1 pre-

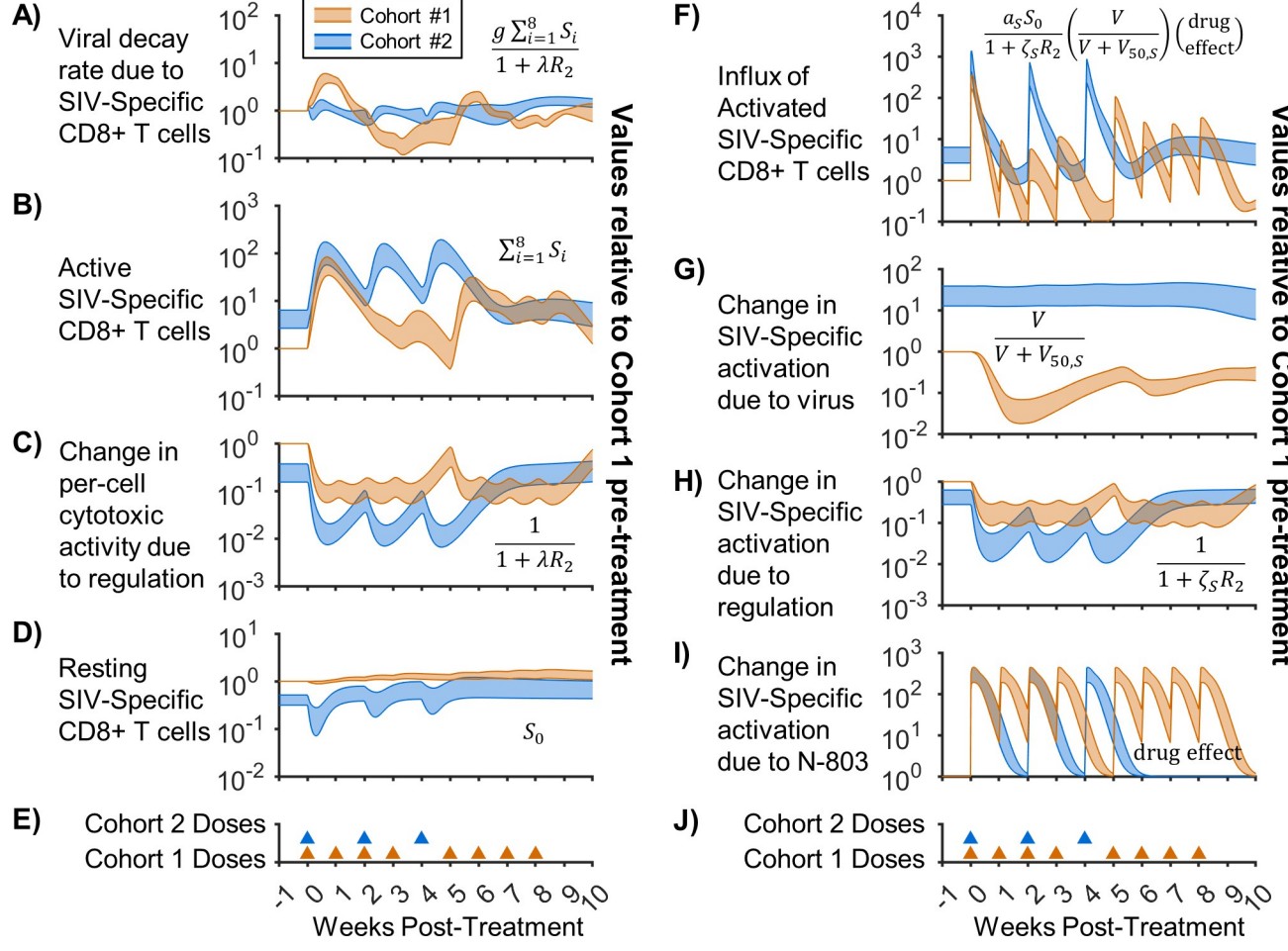

**Fig 5. Factors contributing to viral suppression and CD8$^+$ T cell activation.** Shown is a breakdown of important model terms governing viral suppression and CD8$^+$ T cell activation for Cohort 1 (low viral load, orange) and Cohort 2 (high viral load, blue). Values for each fitted parameter set were normalized to the Cohort 1 pre-treatment baseline for that parameter set, with Bayesian 95% credible intervals of the normalized values shown. Equivalent terms from Eqs 1–11 are shown within each axis. Panel A shows the decay rate applied to the virus by SIV-specific CD8$^+$ T cells (Eq 1), with B & C separating this term into the active SIV-specific CD8$^+$ T cell count and the change in cytotoxicity due to regulatory inhibition. Panel F shows the influx of SIV-specific CD8$^+$ T cells from the resting compartment (Eqs 2 and 3), with panel D showing the resting SIV-specific CD8$^+$ T cells and panels G-I showing the contributions of antigenic stimulation, regulatory inhibition, and N-803 promotion to the activation rate. Regulation in panels C,H represents the effects of immunosuppressive cells and immune checkpoint molecules (Fig 1, Eqs 10 and 11). Panels E,J show the timing of 0.1 mg/kg subcutaneous doses of N-803 for each cohort.

treatment value. Thus, we focused on the relative change of important mechanisms and viral variables with respect to the Cohort 1 pre-treatment baseline.

We first evaluated the impact of SIV-specific CD8+ T cells on the viral load. Fig 5A shows the decay rate of the viremia due to SIV-specific CD8+ T cells in the model, which would be analogous to the percentage of infected cells killed per day. Fig 5B and 5C separate this rate into the contribution of active SIV-specific CD8+ T cell count (5B) and per-cell reduction in killing due to immune inhibition (5C). In Cohort 1, there was a sharp increase in killing when the active SIV-specific CD8+ cell population increases following the first dose (Fig 5B). Although Cohort 2 has similar or greater levels of cytotoxic cells compared to Cohort 1 following dosing (Fig 5B), there was no corresponding increase in CD8+ T cell-dependent viral decay rate (Fig 5A). Granzyme-B must be released from the CD8+ T cell to kill a target cell [70], and this release can be prevented by immune checkpoint molecules [71]. We modeled this by applying inhibitory regulation to the cytotoxic function of CD8+ T cells (Fig 5C). The effect of inhibition was greater for Cohort 2 both before and during N-803 treatment. Since all model constants (e.g. $\lambda$) were shared between cohorts, this difference was due to higher values of the immune regulation variables ($R_{1,2}$, Eqs 10 and 11). Also, the difference in initial regulation between Cohorts 1 & 2 was not set arbitrarily but was rather a function of viral load (see S1 Text for details), thus representing increased immune checkpoint molecule expression with viral load [22,23]. In other words, the model predicts increases in cytotoxic CD8+ T cells in both cohorts, but the cells in Cohort 2 were rendered less effective due to inhibitory factors.

Fig 5F shows the model predicted daily influx of active SIV-specific CD8+ T cells from the resting state. Mathematically, this influx is a function of available antigen (viral load) (Fig 5G), regulatory inhibition (Fig 5H), N-803 promotion (Fig 5I), and the resting cells available for activation (Fig 5D). There were overlapping peaks in activation for both cohorts with the first N-803 dose, with Cohort 2 having higher activation pre-treatment. Further activation in Cohort 2 was limited by the reduced pool of resting cells (Fig 5D). Thus, for Cohort 2, there was a smaller relative benefit from N-803 due to the higher pre-treatment activation. For Cohort 1, the influx of activated cells quickly dropped with continued treatment (Fig 5F), despite strong promotion due to N-803 (Fig 5I). In the model, two factors compromised activation for Cohort 1. First was the reduction of viral load and corresponding loss of antigen (Fig 5G). Second was the increase in regulatory inhibition of activation (Fig 5H), which models potential increases in increases in regulatory T cell frequency and inhibitory molecule expression following N-803 [8]. In the model, regulatory inhibition is generated in response to increased activation, which included the effect of N-803 dosing (Eqs 10). In Cohort 1, the two 2-week break after the fourth dose allowed a partial reset of the regulatory counter-reaction, improving the activation in response to the 5th dose (Fig 5F). The same is shown for the 2-week doses for Cohort 2. This theoretical benefit of 2-week dosing was demonstrated in our previous model [21].

Collectively, these results point to immune regulatory mechanisms (regulatory T cells, checkpoint molecules, etc.) as a reasonable model to explain both the lack of viral suppression following N-803 in high viral load settings (Cohort 2) and the rebound after N-803-induced viral suppression in low viral load settings (Cohort 1).

## The model suggests that extending the IL-15 regimen may lower viral load but not achieve clinical control

We next considered if viral suppression via N-803 can be improved in the high viral load cohort by altering the treatment regimen. Fig 6 shows model predictions of the changes in viral load for the high viral load cohort in response to different dose timings.

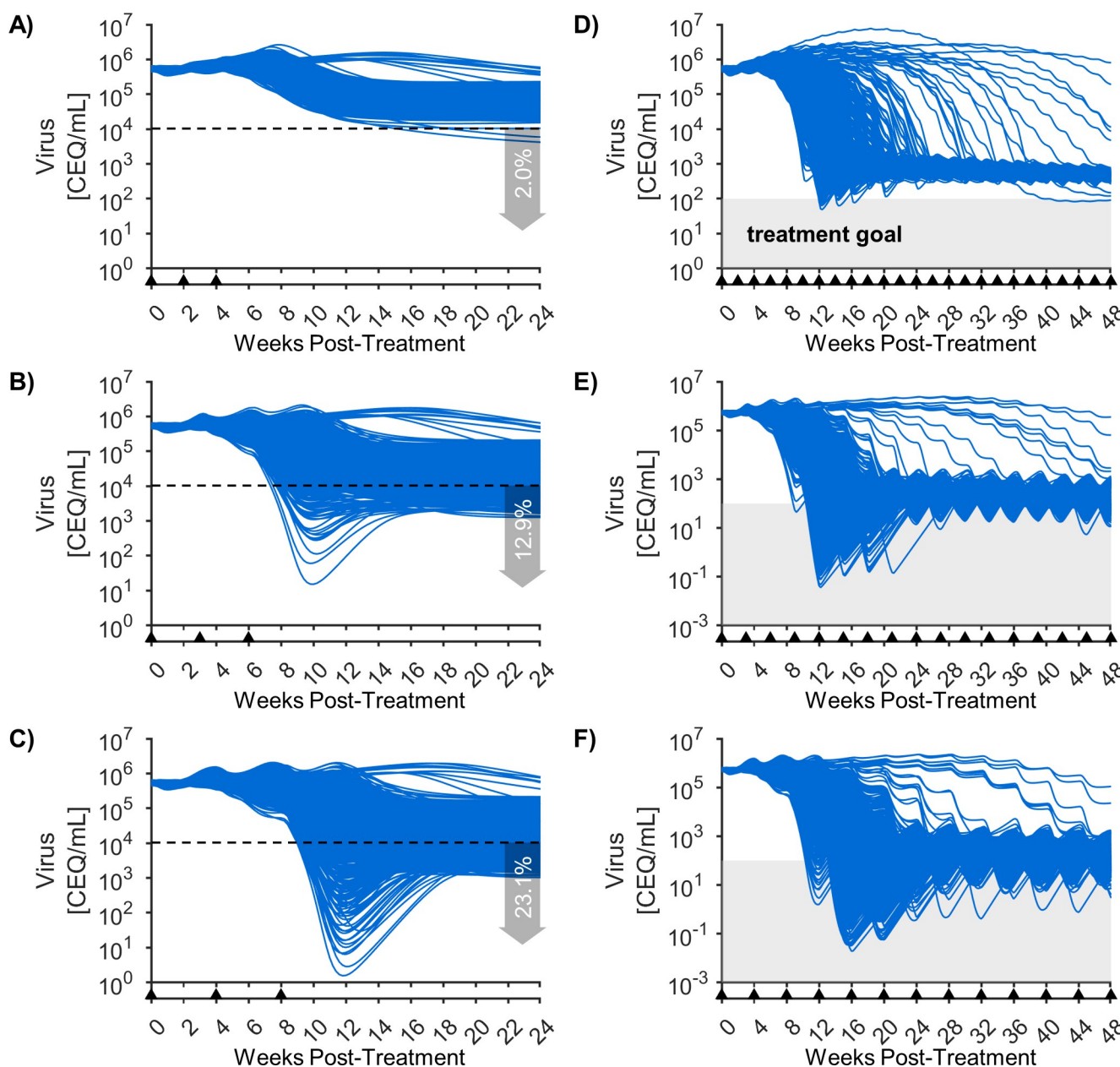

**Fig 6. Simulation of alternative regimens for the high viral load cohort.** Shown are the model predicted responses of the high viral load cohort using 1000 parameter sets (thinning the Bayesian MCMC sample by selecting every 10[th] set). Panels A,B,C show the response to 3 doses given either 2, 3, or 4 weeks apart (the 2-week regimen being the original regimen). Inlayed in these panels are the percentage of the cohort with viral loads <4 log CEQ/mL (marked by a dashed line) at 24 weeks after the first dose. Panels D,E,F show the response to longer 48 week regimens with either 2,3, or 4 weeks between doses. The shaded region shows the limit of detection for common viral assays (100 CEQ/mL), and is considered a treatment goal.

The regimen in our training data consisted of three doses given at 2-week intervals (Fig 6A), and is treated as a control. For this case, model projections after the treatment window show a small fraction of cases (2.0%) where viral load drops close to the pre-treatment set point for the low viral low cohort (<4 log CEQ/mL) by 24 weeks after the first dose is given. The dynamics of the drug-induced regulatory response (Fig 5C and 5H) suggest that less frequent dosing could allow regulatory signals to decrease between doses and increase the efficacy

of each dose. Spreading out the three doses with 3- or 4- week intervals (Fig 6B and 6C) increased the fraction of cases <4 log CEQ/mL to 12.9% and 23.1%, respectively. The underlying model mechanisms leading to this difference in viral suppression are illustrated in S5 Fig. In our model, viral steady-state occurs when the viral growth rate is balanced by the viral suppression due to CD8$^+$ T cell killing of infected cells (Eq 1). The latter is a function of the number of active CD8$^+$ T cells and the regulatory reduction in killing efficacy. At first, treatment pushes the viral load slightly above the initial set point for cohort 2 (S5A Fig). When the drug-induced regulation abates after the last dose (S5C Fig), this elevated viral load boosts CD8$^+$ T cell activation (S5B Fig). This, in turn, pushes viral load back down until the balance between activation and regulation is restored over time. Over very long time periods, the mathematical system either returns to its original steady state viral load or to a new lower viral load (S6 Fig). Spacing out the initial three doses allows for larger swings in virus, CD8$^+$ T cells, and regulation between each dose (S5E–S5G Fig). This increase in momentum perturbs the system sufficiently to push it to a lower steady-state.

We next evaluated if the decline in viral load following three doses would be increased if additional doses were given. Fig 6D–6F show the response of the system to doses given every 2, 3, or 4 weeks for 48 weeks. In all three cases, the virus settles to a consistent cycle within each dose period for most simulated patients by roughly 24 weeks of treatment. A clinical goal for antiretroviral therapy is maintaining viral load below the limit of detection for standard assays (50–100 CEQ/mL) [72]. None of these regimens achieve that consistently. This limitation stems from the viral-dependence of CD8$^+$ T cell activation in our model, which weakens the CD8$^+$ T cell response when viral load is very low (Fig 5F and 5G). These results should be viewed with caution, as they are longer term extrapolations of a model that was fit to a much shorter regimen. However, they at least suggest that an N-803 regimen may be as effective, if not more so, if the time between doses is extended.

## Discussion

Here, we adapted and expanded our previous mechanistic mathematical model [21] to replicate two distinct responses to an IL-15 agonist (N-803) observed in two SIV-infected NHP cohorts. Our simulations of the two cohorts assumed the same model constants for both cohorts but allowed different pre-treatment steady-states. These states being the initial values of model variables representing viral load, CD8$^+$ T cells, and a phenomenological amalgamation of immune inhibitory factors such as regulatory T cells and immune checkpoint molecules. The model was able to capture the CD8$^+$ T cell response to subcutaneous N-803 administration in both cohorts (Fig 3B and 3E), the transient SIV suppression in Cohort 1 (Fig 3A), and the lack of SIV suppression in Cohort 2 (Fig 3D). The model was qualitatively compared to the experimentally measured frequency of cytotoxic cells among SIV-specific CD8$^+$ T cells in both cohorts (Fig 4) and against CD8$^+$ T cells and SIV viral load from independent NHP cohorts (S4 Fig). This illustrates that applying an activator like IL-15 during a state of high immune activation yields a diminished return. In Cohort 2, regulatory inhibition of CD8$^+$ T cell killing (Fig 5C) limited viral suppression by SIV-specific cells (Fig 5A) despite increases in cytotoxic cell counts (Fig 5B). In Cohort 1, loss of antigen stimulation (Fig 5G) combined with regulatory inhibition (Fig 5H), lowered CD8$^+$ T cell activation with repeated N-803 doses (Fig 5F), allowing viral rebound.

While this work demonstrates that the divergent N-803 responses of the two cohorts can be replicated by a mathematical model where the cohorts differed by the initial state alone, there were also additional biological differences between the cohorts (Table 3). Three of the four NHPs in Cohort 1 expressed the *Mamu-B*08* MHC allele [8], which has been associated with

better SIV control [61]. Most notably, Cohort 1 temporarily controlled SIV as part of a previous study [60]. The biological mechanisms behind spontaneous HIV control are still a subject of much research [73,74]. Inherent differences between the cohorts could be represented mathematically by different model constant values, which could allow for theoretical assessment of potential mechanisms of HIV control. Here we illustrated that these inherent biological differences are not mathematically necessary in order to recreate the observed viremia dynamics.

Both this current work and our previous work [21] theoretically demonstrate how immune regulatory mechanisms could be responsible for the lack viral suppression by N-803. Since N-803 seems to be effectively expanding cytolytic SIV-specific cells in both cohorts (Fig 5B), direct reduction of the per-cell killing rate of cytotoxic cells (Fig 5C) is necessary to preclude viral suppression in the mathematical model. Biologically, this reduction in cytotoxic activity could be caused by shielding of infected cells via PD-L1 on these cells, signaling through PD-1 on cytotoxic cells and stopping the cytolytic response [71]. A combination of N-803 with two anti-PD-L1 domains (N-809) was able to promote the cytotoxic function of CD8$^+$ T cells and natural killer cells as well as bind to tumor cell PD-L1, synergistically increasing tumor cell lysis in vitro and in murine models [30,31]. In addition, N-809 reduced the frequency of immunosuppressive regulatory T cells (defined as CD4$^+$ FoxP3$^+$ T cells) in the tumor environment [31]. In a mouse model of chronic lymphocytic choriomeningitis virus, depletion of regulatory T cells (also CD4$^+$ FoxP3$^+$) expanded virus specific CD8$^+$ T cells but did not reduce viral load, possibly due to a simultaneous increase in PD-L1 on target cells [75]. When regulatory T cell depletion and PD-1 blockade were simultaneously applied in the mouse model, viral load was reduced. It is conceivable that combining N-803 with regulatory T cell depletion may be redundant, as both would increase virus-specific CD8$^+$ T cell numbers while reducing their killing efficacy by increasing PD-1/PD-L1 expression.

Similar combinations of immune activators, like N-803, and immune checkpoint inhibitors are being included in multi-drug therapy of cancer. In cancer, chronic inflammation promotes expression of immune checkpoint molecules and their ligands [76,77], which limit the effectiveness of IL-15 therapy [30,31,78]. Inflammation also recruits immunosuppressive cells, such as myeloid-derived suppressor cells (MDSCs) [79], which, in turn, attenuate the effectiveness checkpoint blockade therapy [80,81]. In light of these many factors, increasingly complex therapies are being proposed. A murine tumor model demonstrated the effectiveness of a six-drug combination therapy that included N-803 and an anti-PD-L1 antibody [82]. A recent case study saw complete remission of Merkel cell carcinoma following a combination of N-803, PD-1 blockade, and a chemotherapeutic [83]. Both of these regimens included taxanes, which are anti-mitotic chemotherapy drugs that also promote antigen presentation in cancer cells to enhance targeting by CD8$^+$ T cells [84]. Some chemotherapeutics can also target MDSCs [79,85], and combining N-803 with sunitinib to suppress MDSCs synergistically inhibited melanoma growth and promoted survival in a murine model [86]. Treatment of certain types of tumors may require these multifaceted approaches [82], and mechanistic mathematical models, such as the one developed here, could be developed to inform the integration of these approaches and design regimens that may have improved efficacy.

In HIV, the approach outlined above would be somewhat analogous to combining immune therapy with traditional antiretroviral therapy (ART). ART can reduce PD-1 expression in CD8$^+$ T cells [87,88], and reduce the frequency of CD4$^+$ FoxP3$^+$ regulatory T cells [89,90]. Comparison of CD8$^+$ T cell proliferation and cytotoxicity markers between these two cohorts supported the assertion that N-803 is best used in the context of patients that control viral loads [8]. Though a recent clinical trial demonstrated N-803 can be safely administered to ART-suppressed HIV patients [14], combination of N-803 and ART in an animal cohort did

not reduce the latent SIV reservoir or preclude viral rebound after ART interruption [91]. There is still the potential that the further addition of PD-1 blockade to ART and N-803 combination therapy could allow viral control with less frequent dosing. There are additional mechanisms that would be relevant for a mathematical model to consider application of N-803 to ART-suppressed HIV. First is the capacity of N-803 to reverse latent HIV infections [91,92]. Latency was excluded from this model under the assumption that reactivation of latent SIV would only be a small contribution to unsuppressed viral loads, given the small frequency of latently infected cells in chronic HIV infection [46]. Second, N-803 can induce CD8[+] T cell infiltration into B cell follicles [10], which are sites with dendritic-cell-bound viral reservoirs and limited CD8[+] T cell presence [93].

In summary, we used a mathematical model, informed by in vivo treatment data, to demonstrate that the response to a therapeutic cytokine will depend on the pretreatment immune state and the balance between immune activation and regulation. These results can provide further insights into the application of IL-15-based therapy in the treatment of both cancer and chronic infections.

## Supporting information

**S1 Text. Document containing additional methodological details, results, and discussion, which are all referenced within the article.**
(PDF)

**S1 Fig. Constant and initial condition distributions.** Shown is the Bayesian MCMC sample of values for constants (A) and initial conditions (B) in Eqs 1–11. Solid lines indicate allowed ranges, which are also given in Tables 1 and 2. Constants and initial conditions without ranges are calculated by assuming pre-treatment steady-state or by sampling ratios relative to other parameters. Units are as given in Table 2, except that $g$ is converted to nL/#/d.
(TIF)

**S2 Fig. Comparison of model to mean of experimental data.** This is a repeat of Fig 3, but where the model is instead compared to the mean of the data points, across each cohort, at each time point. The model was calibrated to (A,D) virus in the plasma and (B,E) CD8+ T cells in the peripheral blood from two different Simian Immunodeficiency Virus (SIV) cohorts. The orange and blue shaded regions correspond to the Bayesian 95% credible interval of the mathematical model. The gray shaded region indicates the lower limit of detection for the viral assay (100 CEQ/mL). Panels (C,F) show timing of 0.1 mg/kg subcutaneous doses of N-803.
(TIF)

**S3 Fig. Comparison of fitting to single cohorts and fitting to both cohorts.** The model was calibrated to (A,D) log fold change in virus in the plasma and (B,E) fold change in CD8+ T cells in the peripheral blood from two different Simian Immunodeficiency Virus (SIV) cohorts. Shown are top 10 results (lowest NLL) from the multi-start local-search algorithm for two different scenarios. The orange/blue lines correspond to the scenario in Fig 3, where the same set of model constants is used for both cohorts ("Shared"). The black lines correspond the case where different sets of constants are fitted to each cohort ("Unshared"). The gray shaded region indicates the lower limit of detection for the viral assay (100 CEQ/mL). Panels (C,F) show timing of 0.1 mg/kg subcutaneous doses of N-803. See S1 Text for additional methodological details.
(TIF)

**S4 Fig. Comparison of model to other non-human primate cohorts.** Model predictions are compared to a different SIV NHP cohort [10]. In panel A, SIV-naïve NHPs (n = 4) are given a 6 mg/kg intravenous dose of N-803 at week 0. In panel D, SIV-infected NHPs (n = 4) are given three 6 mg/kg doses spaced one week apart, followed by a 100 mg/kg dose 5 weeks later. Peripheral blood CD8+ T cell counts are shown in both panels (mean and standard deviation) and compared to mathematical model predictions (Bayesian 95% credible interval). Panel C shows the plasma viral load after the 100 mg/kg dose (one data subject compared to model prediction). NHP data was obtained from published figures using Engage Digitizer software. Panels (B,E) show timing and size of intravenous doses of N-803. See S1 Text for additional methodological details and discussion of results.
(TIF)

**S5 Fig. Simulation of alternative regimens for the high viral load cohort.** Shown are the model predicted responses of the high viral load cohort using a subset of 1000 parameter sets from the Bayesian MCMC parameter sample, representing a simulated cohort. Panels A-D and E-H correspond to 3 doses given either 2 or 3 weeks apart, respectively (the 2-week regimen being the original regimen). Panels A,E show the viral load, panels B,F show the total active SIV-specific CD8+ T cells, and panels C,G show the change in cytotoxicity due to regulatory inhibition. Equivalent terms from Eqs 1–11 are shown within each axis. Values for each fitted parameter set were normalized to the Cohort 2 pre-treatment baseline for that parameter set (marked by dashed line. Panels D,H show the timing of 0.1 mg/kg subcutaneous doses of N-803.
(TIF)

**S6 Fig. Long-term projections of viral load.** Shown are long-term model projections after the last N-803 is given to low viral load cohort (panel A) and high viral load cohort (panel B) using a subset of 1000 parameter sets from the Bayesian MCMC parameter sample.
(TIF)

## Acknowledgments

We thank ImmunityBio for supplying the reagent N-803.

## Author Contributions

**Conceptualization:** Jonathan W. Cody, Amy L. Ellis-Connell, Shelby L. O'Connor, Elsje Pienaar.

**Data curation:** Jonathan W. Cody, Amy L. Ellis-Connell.

**Formal analysis:** Jonathan W. Cody.

**Funding acquisition:** Shelby L. O'Connor, Elsje Pienaar.

**Investigation:** Jonathan W. Cody.

**Methodology:** Jonathan W. Cody, Amy L. Ellis-Connell, Shelby L. O'Connor, Elsje Pienaar.

**Project administration:** Elsje Pienaar.

**Resources:** Elsje Pienaar.

**Software:** Jonathan W. Cody.

**Supervision:** Elsje Pienaar.

**Validation:** Jonathan W. Cody.

**Visualization:** Jonathan W. Cody.

**Writing – original draft:** Jonathan W. Cody.

**Writing – review & editing:** Jonathan W. Cody, Amy L. Ellis-Connell, Shelby L. O'Connor, Elsje Pienaar.

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
