## [Decision Letter · Decision Letter 0]

28 Feb 2023

Dear Dr Pienaar,

Thank you very much for submitting your manuscript "Mathematical modeling indicates that regulatory inhibition of CD8+ T cell cytotoxicity can limit efficacy of IL-15 immunotherapy in cases of high pre-treatment SIV viral load" for consideration at PLOS Computational Biology.

As with all papers reviewed by the journal, your manuscript was reviewed by members of the editorial board and by several independent reviewers. In light of the reviews (below this email), we would like to invite the resubmission of a significantly-revised version that takes into account the reviewers' comments.

Please respond carefully to the reviewers' comments, especially the issues around model complexity, number of parameters to be fitted and corresponding interpretation of the results. The model justification should include not just why it matches the biology, but also how it is commensurate with the available data to constrain the model. Likely, it would be beneficial to test sensitivity of your results to the model structure, including appropriate simplified versions. This is clearly a major revision, which should be detailed in your response, and upon which an acceptance or a rejection decision will be made.

We cannot make any decision about publication until we have seen the revised manuscript and your response to the reviewers' comments. Your revised manuscript is also likely to be sent to reviewers for further evaluation.

Sincerely,

Ruy M. Ribeiro

Guest Editor

PLOS Computational Biology

Amber Smith

Section Editor

PLOS Computational Biology

Reviewer's Responses to Questions

**Comments to the Authors:**

Reviewer #1: In the article titled “mathematical modeling indicates that regulatory inhibition of CD8+ T cell cytotoxicity can limit efficacy of IL-15 immunotherapy in cases of high pre-treatment SIV viral load”, the authors built on their previous work to study the driving factor behind the low efficacy of IL-15 treatment for SIV infected non-human primate. As the title suggests, the main finding is that IL-15 treatment is limited due to regulatory inhibition of CD8+ T cells in non-human primates with high pre-treatment viral load. The study is well-motivated and the result is intuitive, well-reasoned and demonstrated. The technical aspects, such as model parametrization, are strong. The entire procedure seems clearly presented and should be reproducible. I only have a few minor comments regarding the manuscript.

1. I find the model somewhat unnecessarily complicated. For example, does including S1 to S8 compartments for the different activated stages of CD8+ T cells necessarily improve model fit, because it does not seem to affect any other aspect of the manuscript? Have the authors tried to simplify the models?

2. While the methods are clear, it is still rather convoluted. I think having a flow-chart figure showing the steps taken for model parametrization would be helpful. Additionally, fitting is done on individual patients simultaneously (Eq. S38-39), so it seems including a plot with the fit and the average of the data would be necessary, especially because the 95% credible interval misses the high and low points due to individual heterogeneity (Fig. 1). Another point is that the 95% credible interval for cohort 2 seems to indicate a decreasing trend near the end – week 8 post treatment - (Fig. 1D), which corresponds to an increasing CD8+ T cells (Fig. 1E). Since the model dynamics is not known analytically, extending the simulation to check whether this declining trend leads to viral extinction is necessary. I bring up this point because of the SIV dynamic (Eq. 1) can be written as V’ = (…)V, so if the terms in (…) is negative, SIV may become extinct.

3. After MCMC to obtain posterior distributions of the initials, are the mean used for subsequent fitting and analysis? I think this information is missing from the manuscript. However, if this is the case, there is another potential issue. The study looks at the “average” of the population to avoid individual heterogeneity; however, for such a complicated models, chaotic behaviors are not ruled out. Thus, infinitesimally small difference in initials can lead to completely different dynamical behavior qualitatively. And because the parameter distribution of the initials are so wide (some vary over 4-5 logs in value), this point should be addressed for the results to be credible.

4. Some minor preferences. I find the term “validated” a bit too strong. Something like “tested” or “supported” maybe more suitable. Also, while the authors frame the differences in the initials (pre-treatment viral load, etc.) “can influence immunotherapeutic efficacy” (abstract), the results show the underlying differences (differences in parameters) are the reason behind this. On this note, please clarify the statement on lines 433-434.

Reviewer #2: The paper uses a mathematical model and two cohort data to determine virus-host differences in subjects that respond to an IL-15 antagonist therapy and the ones who do not. The idea is interesting, but the presentation is very dense and the model needs further justification and explanation. I will detail my suggestions below.

1. The model is very large, with 20 equations for immune activation following drug initiation. It is adapted from previous studies and not completely explained here. That makes it hard to follow. In particular:

a. Why do you need an antigen dependent and antigen independent expansion of T cells?

b. Why do you use 7 classes in the S expansion? Is that number relevant, or fewer classes will suffice?

c. Why do you use an antigen class and not the virus class V for immune activation? You can apply the drug effect directly into the V equation.

d. Why did you not consider a model of viral infection (with target cells and infected cells)?

e. You used quasi-equilibrium assumptions during data fitting. Can you reduce the model taking those into consideration?

2. I have several issues on data fitting. In particular:

a. Can you explain how you fit the population to data on fold changes rather than concentration values.

b. It is not clear which parameters are being fitted (especially in the main manuscript). Please explain that.

c. What are the differences between the three data symbols? It was hard to follow which ones were below limit of detection. List how many data points you use for data fitting, that way one can compare number of parameters being estimated to number of data points.

d. Did you compute an AIC for the model after data fitting?

3. Can you use the results from cohort 1 to explain the low initial data subjects in cohort 2?

4. Can you use the model to design in-silico experiments that can tell you under what conditions the drugs will work for the high initial titer? Such as using different dosing or timings?

5. Minor issues:

a. The use of p and P in the same equation, same with a and A.

b. the sum of S+N should have summation indexes.

c. Add A and H into the variables table.

**Have the authors made all data and (if applicable) computational code underlying the findings in their manuscript fully available?**

Reviewer #1: **No: **The authors refer to other publications for data set. No explicit data set is given.

Reviewer #2: **No: **I did not see any code or data being shared within the manuscript of SI.

PLOS authors have the option to publish the peer review history of their article (what does this mean?). If published, this will include your full peer review and any attached files.

Reviewer #1: No

Reviewer #2: No
---

## [Decision Letter · Decision Letter 1]

13 Jun 2023

Dear Dr Pienaar,

Thank you very much for submitting your revised manuscript "Mathematical modeling indicates that regulatory inhibition of CD8+ T cell cytotoxicity can limit efficacy of IL-15 immunotherapy in cases of high pre-treatment SIV viral load" for consideration at PLOS Computational Biology.

We appreciate your efforts in revising the manuscript and providing a reply to reviewers. However, your innovative way of structuring the reply, aggregating related comments instead of following the order of comments with each comment and the corresponding answer, made it harder for the reviewers to evaluate your revised manuscript. Indeed, one of the reviewers requested explicitly to have each reviewers' comment followed by the reply to be able to evaluate your revision. Could you re-arrange your reply to reviewers in that standard way, and include the actual comments of the reviewer together with the corresponding answers in order?

We apologize for this inconvenience and for taking some time to reach this conclusion. However, our reviewers are extremely busy scientists and we need to help them do their reviews efficiently.

Note that at this time, this was considered a "Minor revision", because the revised manuscript still has to complete peer-review. But if you feel that you have replied to all the comments, this should be a simple reorganization of the reply text.

Sincerely,

Ruy M. Ribeiro

Guest Editor

PLOS Computational Biology

Amber Smith

Section Editor

PLOS Computational Biology

Reviewer's Responses to Questions

**Comments to the Authors:**

The authors did not include the reviewer critiques with their responses and did not address the recommendations one by one. Instead they selected quotes from the reviews that they found relevant and only addressed those. Because of that, a recommendation cannot be made on the updated manuscript.

**Have the authors made all data and (if applicable) computational code underlying the findings in their manuscript fully available?**

Figure Files:

Data Requirements:

Reproducibility:

References:

---

## [Decision Letter · Decision Letter 2]

10 Aug 2023

Dear Dr Pienaar,

Thank you for submitting a version of the ms with detailed response to the reviewers' comments. After the reviewers' evaluation, we are pleased to inform you that your manuscript 'Mathematical modeling indicates that regulatory inhibition of CD8+ T cell cytotoxicity can limit efficacy of IL-15 immunotherapy in cases of high pre-treatment SIV viral load' has been provisionally accepted for publication in PLOS Computational Biology.

Best regards,

Ruy M. Ribeiro

Guest Editor

PLOS Computational Biology

Amber Smith

Section Editor

PLOS Computational Biology

Reviewer's Responses to Questions

**Comments to the Authors:**

Reviewer #2: My comments and suggestions have been addressed and the paper is acceptable for publication.

**Have the authors made all data and (if applicable) computational code underlying the findings in their manuscript fully available?**

Reviewer #2: Yes

PLOS authors have the option to publish the peer review history of their article (what does this mean?). If published, this will include your full peer review and any attached files.

Reviewer #2: No

---

## [Editor Report · Acceptance letter]

18 Aug 2023

PCOMPBIOL-D-23-00038R2 

Mathematical modeling indicates that regulatory inhibition of CD8+ T cell cytotoxicity can limit efficacy of IL-15 immunotherapy in cases of high pre-treatment SIV viral load

Dear Dr Pienaar,

I am pleased to inform you that your manuscript has been formally accepted for publication in PLOS Computational Biology. Your manuscript is now with our production department and you will be notified of the publication date in due course.

With kind regards,

Dorothy Lannert
